# Milk Composition of Free-Ranging Impala (*Aepyceros melampus*) and Tsessebe (*Damaliscus lunatus lunatus*), and Comparison with Other African Bovidae

**DOI:** 10.3390/ani11020516

**Published:** 2021-02-17

**Authors:** Gernot Osthoff, Arnold Hugo, Moses Madende, Lauren Schmidt, Sibusiso Kobeni, Francois Deacon

**Affiliations:** 1Department of Microbial, Biochemical and Food Biotechnology, University of the Free State, Bloemfontein 9300, South Africa; hugoa@ufs.ac.za (A.H.); mosesmadende@gmail.com (M.M.); schmidtlaurenlorraine@gmail.com (L.S.); kobenisbu@gmail.com (S.K.); 2Department of Animal, Wildlife and Grassland Sciences, University of the Free State, Bloemfontein 9300, South Africa; deaconf@ufs.ac.za

**Keywords:** Artiodactyla, tsessebe, impala, milk, protein, fatty acid, lactose, oligosaccharide

## Abstract

**Simple Summary:**

Until now, the milk composition of impala and tsessebe has been unknown. Our study showed that the composition of impala milk was 5.56 ± 1.96% fat, 6.60 ± 0.51% protein, and 4.36 ± 0.94% lactose, and that of tsessebe milk was 8.44 ± 3.19%, 5.15 ± 0.49%, and 6.10 ± 3.85%, respectively. The fatty acid composition and protein properties also differed. The data of these two species were subjected to an interspecies comparison with 13 other antelope species by statistical methods. This showed that the milk of tsessebe is similar to that of its relatives of the Alcelaphinae sub-family. Although the impala is a close relative of the Alcelaphinae, its milk composition finds comparison with a different sub-class, the Hippotraginae. The information contributes to the phylogenetic properties of milk and milk evolution.

**Abstract:**

The major nutrient and fatty acid composition of the milk of impala and tsessebe is reported and compared with other Bovidae and species. The proximate composition of impala milk was 5.56 ± 1.96% fat, 6.60 ± 0.51% protein, and 4.36 ± 0.94% lactose, and that of tsessebe milk was 8.44 ± 3.19%, 5.15 ± 0.49%, and 6.10 ± 3.85%, respectively. The high protein content of impala milk accounted for 42% of gross energy, which is typical for African Bovids that use a “hider” postnatal care system, compared to the 25% of the tsessebe, a “follower”. Electrophoresis showed that the molecular size and surface charge of the tsessebe caseins resembled that of other Alcelaphinae members, while that of the impala resembled that of Hippotraginae. The milk composition of these two species was compared by statistical methods with 13 other species representing eight suborders, families, or subfamilies of African Artiodactyla. This showed that the tsessebe milk resembled that of four other species of the Alcelaphinae sub-family and that the milk of this sub-family differs from other Artiodactyla by its specific margins of nutrient contents and milk fat with a high content of medium-length fatty acids (C8–C12) above 17% of the total fatty acids.

## 1. Introduction

Insight into the nutritional and biochemical properties of milk synthesis is difficult to investigate in a single species. However, this may be overcome by comparative studies between species. Milk of ruminants, especially the commercially exploited animals, has been studied extensively. These include the cow (*Bos taurus*), water buffalo (*Bubalus bubalis*), yak (*Bos grunniens*), sheep (*Ovis aries*), goat (*Capra hircus*), camel (*Camelus bactrianus*) [1,2,3,4], and reindeer (*Rangifer tarandus*) [5]. Information of milk from other species is on the increase, e.g., African buffalo (*Syncerus caffer*) [6], blesbok (*Damaliscus dorcas phillipsi*), blue wildebeest (*Connochaetes gnou*) and black wildebeest (*Connochaetes taurinus taurinus*) [7], bongo (*Tragelaphus eurycerus*) [8], kudu (*Tragelaphus streticeros*) and oryx (*Oryx gazella*) [9], springbok (*Antidorcas marsupialis*) [10], sable antelope (*Hippotragus niger*) [11], and giraffe (*Giraffa camelopardalis*) and red hartebeest (*Alcelaphus buselaphus*) [12]. The greatest drawback in the research of milk of exotic and wild species is to obtain statistically representative numbers of samples, because most of the time milk is collected by chance during veterinarian and management procedures. Furthermore, on the basis of the knowledge obtained from the domestic species, conditions that may affect milk composition should be controlled as far as possible, such as equal nutrition of all sampled animals and collection at around mid-lactation [1,2,3,4]. Hence, progress is in small steps.

The milk fatty acid composition differs between species. In foregut fermenters, the ingested fatty acids are changed by fermentation, while lactate, acetate, and 3-hydroxybutyrate, the metabolites from the fermentation of carbohydrates by the rumen bacteria, serve as substrates for de novo synthesis of medium-chain fatty acids. The length of the de novo-synthesized fatty acids depends on the specificity of the enzymes in the synthesis pathway, such as the fatty acid synthase and thioesterase, which together produce fatty acids with chain lengths of up to 16 carbons [13].

In non-ruminants, phylogenetic factors may contribute to the differences of milk fat composition [7,14,15]. Amongst ruminants, this is not clear yet [12], although it is known that milk fat species of the Caprinae sub-family, specifically sheep (*Ovis aries*) [16] and goat (*Capra hircus*) [17,18,19], may contain around 10–19% medium-chain (C8 to C12) fatty acids, compared to less than 10% in other ruminants. The blesbok, red hartebeest, and blue and black wildebeest (subfamily Alcelaphinae) contain 20–35% [6,12]. Amongst the Bovini tribe of the Bovinae, the proximate and fatty acid composition of the domesticated [1] and indigenous African cows [20], African buffalo [6], yak [21], and buffalo [22] differ. The greatest differences of fatty acids were shown to be C16:0, C18:0, and C18:1 [6,12,23].

Regarding the medium chain fatty acid composition in milks, the greatest differences observed were the C8 to C12 acids [24]. The C14:0 content is very constant at below 10%. In general, only milk of ruminants contain above 10% of C14:0, which indicates the chain length specificity of their thioesterase. Contents above 16% have only been reported in blesbok and blue wildebeest [7], blackbuck antelope (*Antilope cervicapra*) [25], and gazelle (*Gazella granti*) [26].

The milk proteins of all ruminants consist of αs1-, αs2-, β-, and κ-caseins, and whey proteins, of which α-lactalbumin and β-lactoglobulin are the major representatives [27]. The concentration of each of these proteins and their relative proportions may differ between species. The genes of these also differ between species, which results in differences in amino acid sequence.

Lastly, in terms of the sugar composition, lactose is the most important carbohydrate in the milk of almost all placental mammals, serving as a source of energy [1]. Lactose is also the main osmole of milk and its production is associated with the movement of water into the secretory vesicle [28]. Oligosaccharides often make up a portion of the sugars. The oligosaccharides are not an energy source but have a protective role in the intestine as pre-biotics for bifidobacteria and lactobacilli [29]. They are also colonic anti-inflammation factors [30] and play a role in neonate brain development [31]. In ruminants, the oligosaccharide content is normally low, between 0 and 4.2 g/kg [9,11,13].

Most reports on inter-species comparison of milk use data from different publications. Although with some success, concerns were raised about the compatibility of data of inter-laboratory origin [24,32,33]. In our own publications in which only two to four species were compared, statistically significant differences in the milk parameters were observed, and in a few cases, the phylogenetic differences between two to four species could be speculated [6,7,9,10,11,20]. When the milk composition of more species from different taxa became available, we were able to extend the comparison by a statistical approach and described phylogenetic differences between several taxa of African ruminants [12].

Impala (*Aepyceros melampus*) occur in the eastern woodland parts of Africa from northern Kenya south to the KwaZulu-Natal region of South Africa, extending westwards to the extreme southern parts of Angola. They both graze and browse, depending on availability of food and season. In the region under study, the lambs are born during December and January, within four to five weeks. The young are hidden for a day or two before joining their mother with the herd. The nursing period is five weeks [34].

Tsessebe (*Damaliscus lunatus lunatus*) have a wide, scattered, and discontinuous distribution from Senegal to eastern Ethiopia and southwards to the Mpumalanga region of South Africa. They are exclusively grazers. In the southern distribution range, the calves are born in October and November. Similar to the other *Alcelaphinae* species, tsessebe young immediately follow the mother, grazing starts at two months of age, while nursing is continued to eight months [34].

In the current study, the milk composition of impala and tsessebe is reported for the first time. This is compared with data of closely related species, as well as 13 other species from 8 African Artiodactyla subfamilies by a statistical comparison, specifically with regard to the lactose, oligosaccharides, proteins, non-protein nitrogen (NPN), fat, and fatty acids, in order to describe phylogenetic differences of milk composition between families and sub-families of Artiodactyla.

## 2. Materials and Methods

### 2.1. Animals and Sample Collection

Milk was obtained from 4 impala of the Koppies Dam Nature Reserve, and 6 impala of the farm Helpmekaar, district Ventersburg, Free State Province. The impalas sampled at Koppies Dam were approximately 10 days postpartum, and those of Helpmekaar at 3 and 7 weeks after the first lambs were born. All lambs were born within a 5-week period. The lactation stage of the tsessebe was approximately 3–5 months. Milk from 3 tsessebe was obtained from the farm Holhoek, district Standerton, Mphumalanga Province. The animals roamed on natural vegetation. The animals were sedated for management purposes, with M99. Milk was obtained by palpation of the teats with sustained pressure on the udder. Milk-letting agents were not administered. Teats were milked out to obtain representative samples, producing 0.5–15 mL. Milk from separate teats of impala was obtained, while pooled samples were collected from the tsessebe. Milk was held on ice while in the field, frozen within 2 h, and kept frozen until analyzed. The milk was thawed in a water bath at 39 °C and mixed by swirling in preparation for analysis.

### 2.2. Determination Water Content

Water content was determined by gravimetry. Approximately 200 µL milk was weighed, dried in a forced convection drying oven for 2–3 h at 100 °C, and re-weighed [35].

### 2.3. Protein Analysis

The crude protein (CP) content of approximately 100 µg milk was determined by the Dumas method using the Dumas method [36]. A conversion factor of 6.38 was used to convert nitrogen (N) content to protein content. NPN (non-protein nitrogen) and whey proteins were obtained by selective precipitation with trichloroacetic acid or acidification with hydrochloric acid according to the method of Igarashi [37], and the nitrogen and protein content of each determined as above. Milk proteins were separated by electrophoresis on a Mighty Small miniature slab gel electrophoresis unit SE 260 (Hoefer Scientific Instruments. Holliston, MA, USA). Milk samples were diluted 1:10 with stacking gel buffer that contained 2–5% sucrose, with bromophenol blue as tracking dye. Sample volumes of 5 μL milk were loaded in the wells in the slab gel. Identification of protein bands was based on comparative electrophoretic mobility of the major proteins from cow and sheep on Urea-PAGE.

### 2.4. Lipid Analysis

Quantitative extraction of total fat was performed with chloroform and methanol in a ratio of 2:1 (*v*/*v*) [38]. Total extractable fat content was determined by weight and expressed as gram fat/100 g milk. Transesterification of fatty acids to form methyl esters (FAME) was performed with 0.5 N NaOH in methanol and 14% boron trifluoride in methanol [39]. The FAME were quantified with a Varian 430 GC, with a flame ionization detector and a fused silica capillary column, Chrompack CPSIL 88 (100 m length, 0.25 mm ID, 0.2 μm film thickness), and column temperature of 40–230 °C (hold 2 min; 4 °C/min; hold 10 min). The FAME in hexane (1 μL) was injected into the column by a Varian 4800 Autosampler, (Varian Inc. Walnut Creek, CA, USA) with a split ratio of 100:1. The injection port and detector were maintained at 250 °C. Hydrogen was used as carrier gas at 45 psi with nitrogen as makeup gas. Chromatograms were recorded by Varian Star Chromatography Software (Version 6.41). Hendecanoic acid (C11:0) was used as internal standard, after it was established that it was not detected in the samples under study. Identification of FAME was by comparison of the relative retention times of FAME peaks from samples with those of standards obtained from Supelco (Supelco 37 Component FAME Mix 47885-U with addition of C18:1c7, C18:2c9t11, C19:0, C22:5).

### 2.5. Carbohydrate Analysis

Carbohydrates were determined by high-performance liquid chromatography with a Waters Breeze system with Biorad Aminex 42C (300 × 7.8) mm (Pall Life Sciences, Ann Arbor, MI, USA) and Waters Sugar Pak 1 (300 × 7.8) mm (Microsep, Johannesburg, South Africa) columns at 84 °C with a differential refractive detector. The mobile phase was de-ionized water eluted at 0.6 mL/min. Samples were de-fatted and de-proteinized with Ultrafree-CL (UFC4 LCC 25) filter devices (Millipore, Merck, Johannesburg, South Africa) centrifuged at 3000× *g*. Samples of 10 μL were injected and quantified with maltotriose, lactose, glucose, and galactose as standards.

### 2.6. Determination of Energy

Gross energy (GE; kcal/g milk) was calculated using the following formula: GE = (9.11 kcal/g * % fat + 5.86 kcal/g * % protein + 3.95 kcal/g * % carbohydrate)/100 [40].

### 2.7. Statistical Analysis

For a phylogenetic comparison of the two species under study with other African Artiodactyla, we incorporated the data of springbok, blue wildebeest, black wildebeest, blesbok, red hartebeest, sable antelope, mountain reedbuck, indigenous African cattle, African buffalo, gemsbok, eland, kudu, and giraffe [6,7,10,11,12,20]. Where necessary, previously published data were re-calculated to the same units used here, i.e., g/100 g milk for nutrients and percentage of total fatty acids. For sable antelope, springbok, gemsbok, and giraffe, respectively 2, 4, 10, and 30 additional milk samples were analyzed since publication, which were included in the current study to increase the representative numbers. In these cases, the animals roamed on locations that differed from that of the animals described in the publications. The diet might possibly have been affected by differences in vegetation types of the respective areas. The potential effect on milk composition is incorporated in the discussion.

Significant differences between means among species were determined by analysis of variance (ANOVA) and multiple comparisons between species by the Tukey–Kramer test at α = 0.05 [41]. Principal component analysis (PCA) was used to visualize variables in a two-dimensional space by Varimax rotation. Hierarchial clustering was used to perform phylogenetic comparisons and to construct dendrograms [41]. Twenty-four fatty acids were included in the statistical analysis. Those not detected (ND) or that only occurred in individual animals at less than 0.1% (C11:0, C15:1c10, C18:3c6,9,12, C21:0, C23:0, C24:0, and all the unsaturated fatty acids of C20–C24 length) were excluded.

## 3. Results

The milk proximate composition of the impalas and tsessebes is shown in Table 1 and the milk fatty acid composition in Table 2. The proximate composition of bovine [1,42], ovine [43], caprine [44], and impala milk [45] is included for comparison.

The nutrient composition of impala milk obtained here is very different compared to that reported for the latter [45]. It should be mentioned that the performance of the protein analysis and fat extraction methods applied in the current study were shown by others to produce erroneous results compared to reference methods, specifically in the analysis of milk with a high fat and oligosaccharide content, such as from marine mammals [46]. The milk fat and oligosaccharide content of impala and tsessebe are in the same order of magnitude as in cow’s milk. It was therefore assumed that the analytical techniques were not affected to the same extent as for that of marine mammals. Our own coefficients of variance (cv) for the analysis methods of these parameters in cow’s milk (n = 11) were 2.73% for fat, 2.15% for nitrogen analysis, 2.31% for lactose, and 2.15–3.17% for the individual fatty acids.

In Figure 1, the electrophoretograms of all three tsessebe and two impala as representatives of the total of 20 animals, are shown. The protein bands of the antelope milk showed similar migration sequences to the proteins of the domestic species, cow, and sheep, however, at different distances of migration. Because the proteins of the two antelope species were run on separate gels with either cow or sheep milk, we carried out interpretation by comparison of calculated Rf values (migration distance) of the caseins relative to the bovine and ovine proteins. Electrophoretograms of earlier studies were consulted for this purpose [6,7,10,11,12,20]. Only the caseins were compared, because the presence of the whey proteins may sometimes be too low to be visible as distinct electrophoretic bands. On the basis of the Rf values, we found that the migration distances in Figure 1 were very close to what would be observed if all were run on a single gel.

## 4. Discussion

### 4.1. Proteins

The 5.45 ± 0.49% protein content of the tsessebe milk is comparable with that of other ruminants [12,32], while the 6.60 ± 0.51 of the impala is on the high side of the scale, which has been connected with antelope that hide their offspring for extended periods during the day [8]. The whey to casein ratios of impala milk proteins was 1:3.5. That that of the tsessebe was 1:8.2, and was comparable to that of other Alcelaphinae species, the wildebeest and specifically the blesbok [7], but not with that of the red hartebeest [12].

Comparison of the protein bands, specifically the caseins, of the impala and tsessebe with each other, as well as other ruminants, was performed by Rf values of the electrophoretic bands taken from earlier work. In Figure 1, it can be seen that the κ- and β-caseins of the tsessebe migrated further than those of the ovine milk, and according to the Rf values, very similar to that of cow’s milk. The α-caseins migrated further than the ovine but shorter than the bovine equivalent. Comparison with earlier work showed that the caseins of the tsessebe migrated at almost equal distances to those of the red hartebeest [12], blesbok, and black and blue wildebeest [7]. This indicated that the sizes and surface charges of the milk proteins of the Alcelaphinae family were conserved. The κ- and β-caseins of the impala migrated shorter than the bovine and ovine equivalents, according to the Rf values. The α-caseins of impala migrated further than the ovine but shorter than the bovine equivalent. Comparison with earlier work showed that the caseins of impala showed almost equal migration distances to that of the sable antelope [11], gemsbok, and scimitar oryx [9] of the Hippotraginae family. Although the milk proteins of only a few Bovidae families, specifically the Antilopinae [12], Bovinae [6], Giraffidae, Reduncinae [12], and Caprinae (goat and sheep as references) had been investigated, distinct electrophoretic patterns were observed for each family. The closest relatives to the impala are the Hippotraginae and Alcelaphinae families [47]. It is interesting that the milk proteins of these two families, according to electrophoretic migration, are very different, and that the electrophoretic properties of impala milk proteins should resemble that of the Hippotraginae.

### 4.2. Carbohydrates

The lactose content of 4.36 ± 0.94% of impala is similar to that of the domesticated species as well as other ruminants, while the 6.10 ± 3.85 of tsessebe milk seems to be higher than that of most ruminants [12,32]. No monosaccharides were observed in impala milk, while approximately 0.08% glucose was observed in tsessebe milk. The presence of glucose was also noted in the milk of other Alcelaphinae species, the wildebeest and specifically the blesbok [7], but not in that of the red hartebeest [12].

### 4.3. Lipids

The fat content of 8.44 ± 3.13 of the tsessebe milk is high and comparable with the 6–12% of other Alcelaphinae [7,12]. The fat content of 5.56 ± 1.96% of the impala milk is low when compared to other ruminants in general [31], and African antelope specifically [12]. 

The lipid fractions of the milk of tsessebe (Table 2) contain high amounts of saturated fatty acids (SFA), 80.24 ± 3.89%, mainly due to the high content of the fatty acids of C6–C12 length. It contains 26.9% C6–C12 length combined, comparable to that of its Alcelaphinae cousins, the black and blue wildebeest and blesbok [7]. The milk fat of the impala contains 71.31 ± 2.584% SFA and only 5.16% of the saturated fatty acids of C6–C12 length combined, which brings it in line with that of the springbok (*Antelopinae*) [10] and the Bovinae, cow [20], African buffalo [6], kudu, and eland [9].

The milk fatty acid content of mammals is dependent on the dietary content thereof. In ruminants, other nutritients, specifically roughage, may also play a role [48,49,50]. The impala in this study lived on Dry Cymbopogon-Themeda veld and the tsessebe on Soweto Highland Grassland (a sub-type of Cymbopogon-Themeda veld) [51,52]. The impala is both browser and grazer, depending on availability, and *Cynodon dactylon* and *Themeda triandra* form a large part of its diet [34]. Nutritional information on the tsessebe is only available for its natural range and not for the grassland region under study [34]. However, since Soweto Highland Grassland also contains a high proportion of *Themeda triandra* and *Cynodon dactylon* [51,52], and these are of the major grass species consumed by other Alcelenaphinae species [34], it may be assumed that the tsessebe also consume them. Chilliard et al. [48] reported that the type of grass had no nominal effect on cow’s milk fatty acid composition, probably because the fatty acid composition does not vary much between grass types [48,50]. Freshness and maturity of grass was shown to have a small effect, specifically on content of conjugated linoleic acid, while drastic changes in milk fatty acid composition was shown to depend on dietary supplementation with seeds, such as linseed or maize, because of the higher fat content [53,54]. Foliage as fodder supplement serves to increase the protein content [48,49]. It may therefore be accepted that the diet of the free ranging impala and tsessebe would not affect milk fatty acid composition to a great extent because they grazed on the same grassland as the species they are compared with. 

### 4.4. Energy

Milk energy density is an indication of the ecology and life-history strategy of a species. Extended periods between nursing may cause milk to be highly concentrated. The mechanism is due to a downregulation of the lactose synthesis from infrequent infant suckling, which results in milk with a high fat and low sugar and protein content, which conserves water, glucose, and protein for the mother [2]. Petzinger et al. [8] showed that a “hider” strategy is observed in bovids for which the milk protein supplies a high proportion (33%) of milk gross energy, such as the tribe Tragelaphini [9]. The impala hide their offspring for the first few days and then stay in the herd, not always close to the mother. During this time, they suckle only a few times per day [34]. The tsessebe offspring follow the mother shortly after birth [34]. The percentage gross energy provided by the milk proteins is approximately 39% of the total energy for the impala and 25.0% for the tsessebe, which is in agreement with the “hider” strategy proposed by Petzinger et al. [8].

### 4.5. Statistical Analysis and Interspecies Comparison

The above interpretations of the data were based on comparisons with other species. In order to validate them, the same statistical analysis was implemented, which provided valuable insights in milk of Artiodactyla of previous work [12]. The current data were compared with the data of all Artiodactyla species studied thus far in our laboratory. The species and the number of samples (n/N = number of milk samples/number of animals) represented were impala (20/10) (family *Aepycerotina*); giraffe (34/13) (family *Giraffidae*); indigenous African cattle (42/42) and African buffalo (4/4) (subfamily *Bovinae*, tribe Bovini); eland (14/7) and kudu (1/1) (subfamily *Bovinae*, tribe Tragelaphini); springbok (9/5) (subfamily *Antilopinae*); Southern reedbuck (1/1) (subfamily *Reduncinae*); sable antelope (4/4) and gemsbok (30/15) (subfamily *Hippotraginae*); and tsessebe (3/3), red hartebeest (4/4), blesbok (7/4), blue wildebeest (10/5), and black wildebeest (6/4) (subfamily *Alcelaphinae*). The parameters used were the milk content of lactose, oligosaccharides, protein, NPN, fat, saturated fatty acids (SFA), monounsaturated fatty acid (MUFA), polyunsaturated fatty acid (PUFA), and 24 individual fatty acids.

The statistical analyses were carried out in a progressive way per species, by ANOVA, as was done in earlier work [9,20], as well as with PCAs and a dendrogram [12]. The PCA of fat, protein, NPN, lactose, and oligosaccharides; PCA of SFA, MUFA, and PUFA; PCA of content of fatty acids; and PCA of all the components combined were similar to that observed previously with regards to the clustering of the data of the species according to their taxa. A clear separation of the Bovinae from the other families—*Alcelaphinae*, *Antilopinae*, *Bovinae*, *Giraffidae*, *Hippotraginae*, and *Reduncinae*—was noted [12]. Therefore, in the current study, the data of impala and tsessebe milk were compared with that of the other species in a dendrogram only (Figure 2). 

In the dendrogram, the ruminants were divided into two main clusters at a Euclidian distance of 18. Broadly, the cluster on the right was characterized by milk with a high fat content of 6–12%, lactose content of 3–6%, and protein content of 4–6%. The fat consists of a high content of saturated fatty acids (>75% of total fatty acids) and combined medium-chain fatty acid (8–14 carbons) above 25%, and less than 25% 18 carbon-length fatty acids. Within this cluster, the blue and black wildebeest form a smaller cluster at a Euclidian distance of 6. Their milk contains a lower protein content and higher fat content compared to the tsessebe, blesbok, and red hartebeest, but a similar fatty acid composition. It should be noted that the new milk data placed the tsessebe in the *Alcelaphinae* cluster together with the four cousin species that were shown to cluster in previous work [12]. The tesessebe milk was very similar to that of the other *Alcelaphinae* in general, and to that of the blesbok specifically. The milk composition of the whole subfamily differs from most other ruminants.

The milk composition of the species in the cluster on the left could be defined by 4–15% fat, 3–6% lactose, and 3–9% protein content. The contents overlap with that of the *Alcelaphinae*, however, the exchange amongst the three macronutrients, i.e., low lactose or protein content for high fat, is within larger margins than observed for the *Alcelaphinae*. This cluster is also distinguished by less than 75% saturated fatty acids, and the medium-chain fatty acids are exchanged for long-chain acids, specifically those of 16–18 carbon lengths and their unsaturated forms. Other extremes are also observed in this cluster, such as the springbok milk having the highest fat content of approximately 14.5% [10] and the mountain reedbuck with the lowest lactose content of 3.4% [12]. Although the milk of *Bovinae* representatives differ with regard to protein and fat contents, at least three of them—indigenous African cattle, eland, and African buffalo—are grouped together due to a similar lactose content of approximately 4%, low saturated fatty acid composition of around 64%, as well as a general similarity of fatty acid composition [6,9,20]. A finer grouping within this cluster is not very accurate, because it lies below the truncation line at a Euclidian distance of 4.18. This may imply that individual uniqueness of composition may play a larger role in describing the milk nutrient composition of each species.

## 5. Conclusions

The milk composition of impala and tsessebe was described. By statistical comparison, these analyses showed that the milk composition of the *Alcelaphinae* differs from species of seven other taxonomic groups of African *Artiodactyla*. Although the impala is a close relative of the *Alcelaphinae*, its milk composition is not comparable with their members. The phylogenetic differences could, in part, be described by biochemical and genetic properties, specifically regarding the synthesis of fatty acids. Phylogenetic properties are very complex and are therefore not restricted to the macro-nutrients. Milk data from more species, as well as representatives of other taxonomic groups, together with additional nutrient parameters, will be needed to refine the phylogenetic relationship of ruminant milk.

## Figures and Tables

**Figure 1 animals-11-00516-f001:**
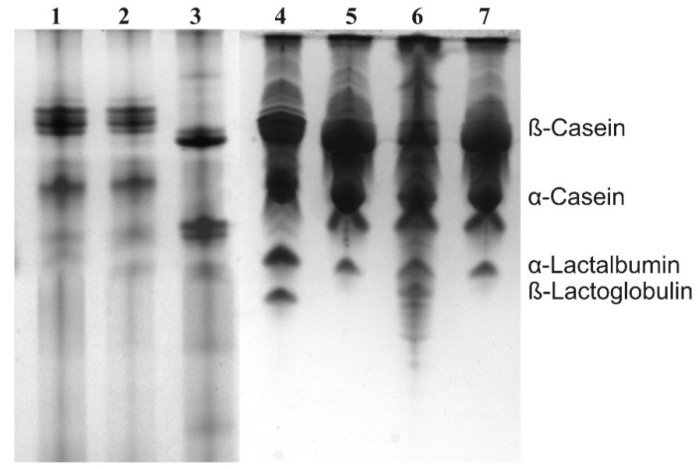
Electrophoretograms of milk from impala (lanes 1 and 2), cow (lane 3), sheep (lane 4), and tsessebe (lanes 5–7).

**Figure 2 animals-11-00516-f002:**
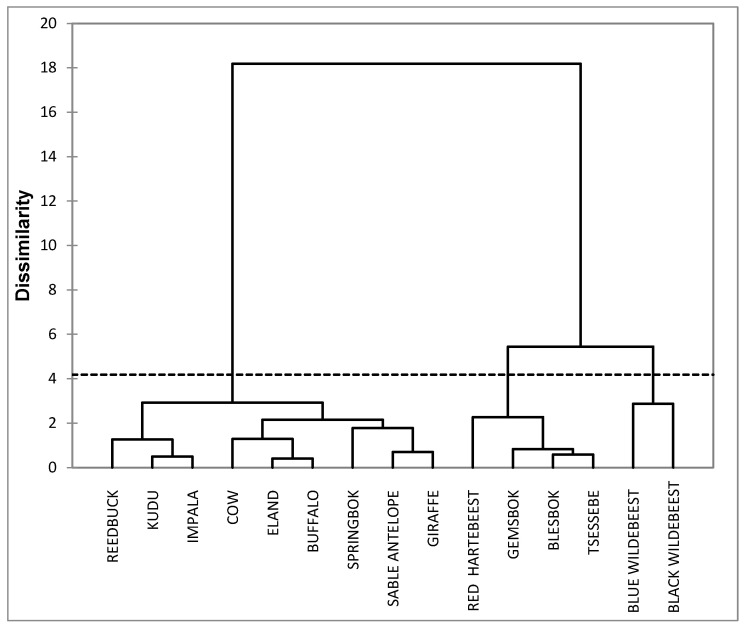
Dendrogram of 15 Ruminantia species with regard to milk content of protein, non-protein nitrogen, lactose, oligosaccharides, fat, and fatty acids.

**Table 1 animals-11-00516-t001:** Proximate composition of the milk of impala and tsessebe compared to domestic animals.

Nutrient (g/100 g Milk)	Impala	Tsessebe	Sheep [43]	Goat [44]	Cow [1]	Impala [45]
n/N	20/10	3/3				
Moisture	85.37 ± 2.28	82.73 ± 1.76	83.2	87.2	87.1	-
Fat	5.56 ± 1.96	8.44 ± 3.19	7.0	5.2	3.9	20.4
FFDM	9.07 ± 1.41	8.82 ± 1.49	-	-	-	-
Protein	6.60 ± 0.51	5.15 ± 0.49	5.3	3.56	3.27	10.8
Whey	1.43 ± 0.60	0.56 ± 0.17	1.04	0.76	0.63	-
Casein	5.09 ± 0.64	4.59 ±0.34	4.26	2.80	2.6	-
NPN	0.09 ± 0.04	0.08 ± 0.02	-	-	-	-
Lactose	4.36 ± 0.94	6.10 ± 3.85	4.30	4.20	4.8	2.4
Galctose	ND	0.08 ± 0.07	-	-	0.05	-
Oligosaccharide	ND	ND	-	-	-	-
Gross energy (kcal/g)	36.63 ± 4.51	45.17 ± 5.44	-	-	-	-

n = number of milk samples; N = number of individual animals; ND = not detected; FFDM = fat-free dry matter; NPN = non-protein nitrogen.

**Table 2 animals-11-00516-t002:** Milk fatty acid composition and fatty acid ratios of impala and tsessebe.

Species			Impala	Tsessebe
n/N			20/10	3/3
FAME (% of Total Fatty Acids)				
Butyric	C4:0		1.23 ± 0.37	0.79 ± 0.04
Caproic	C6:0		0.46 ± 0.19	1.89 ± 0.14
Caprylic	C8:0		0.19 ± 0.31	9.53 ± 0.22
Capric	C10:0		0.96 ± 0.34	11.49 ± 1.15
Hendecanoic	C11:0		ND	ND
Lauric	C12:0		3.55 ± 0.85	3.98 ± 0.23
Tridecoic	C13:0		0.06 ± 0.04	0.06 ± 0.01
Myristic	C14:0		13.95 ± 2.35	14.79 ± 0.9
Myristoleic	C14:1c9		0.15 ± 0.10	0.24 ± 0.03
Pentadecylic	C15:0		2.61 ± 1.67	1.15 ± 0.1
Pentadecenoic	C15:1c10		0.02 ± 0.02	0 ± 0
Palmitic	C16:0		27.89 ± 1.09	29.45 ± 2.87
Palmitoleic	C16:1c9		0.48 ± 0.23	0.55 ± 0.07
Margaric	C17:0		1.09 ± 0.41	0.76 ± 0.14
Heptadecenoic	C17:1c10		0.11 ± 0.11	0.08 ± 0.02
Stearic acid	C18:0		18.65 ± 3.15	11.78 ± 1.47
Elaidic	C18:1t9		0.31 ± 0.43	0.16 ± 0.04
Oleic	C18:1c9		21.45 ± 2.13	15.69 ± 3.3
Vaccenic	C18:1c7		2.08 ± 1.00	0 ± 0
Linolelaidic	C18:2t9,12 (n-6)		0.55 ± 0.34	0 ± 0
Linoleic	C18:2c9,12 (n-6)		1.76 ± 0.55	1.54 ± 0.72
Conjugated linoleic acid	C18:2c9t11(n-6)		0.25 ± 0.07	0.35 ± 0.15
α-Linolenic	C18:3c9,12,15 (n-3)		0.63 ± 0.35	0.35 ± 0.15
γ-Linolenic	C18:3c6,9,12 (n-6)		0.55 ± 0.36	ND
Nonadecanoic	C19:0		0.16 ± 0.08	ND
Arachidic	C20:0		0.34 ± 0.17	0.46 ± 0.05
Eicosenoic	C20:1c11		0.01 ± 0.02	0.04 ± 0.01
Eicosadienoic	C20:2c11,14 (n-6)		0.01 ± 0.01	ND
Eicosatrienoic	C20:3c11,14,17 (n-3)		0.11 ± 0.12	ND
Eicosatrienoic	C20:3c8,11,14 (n-6)		0.04 ± 0.05	0.24 ± 0.03
Arachidonic	C20:4c5,8,11,14 (n-6)		0.13 ± 0.09	0.03 ± 0.02
Eicosopentaenoic	C20:5c5,8,11,14,17 (n-3)		0.03 ± 0.04	0 ± 0
Heneicosanoic	C21:0		0.05 ± 0.04	0.15 ± 0.02
Behenic	C22:0		ND	ND
Erucic	C22:1c13		0.03 ± 0.04	ND
Docosadienoic	C22:2c13,16 (n-6)		ND	0.05 ± 0.03
Docosapentaenoic	C22:5c7,10,13,16,19 (n-3)		0.03 ± 0.04	ND
Docosahexanoic	C22:6c4,7,10,13,16,19 (n-3)		ND	ND
Tricosanoic	C23:0		0.05 ± 0.05	0.16 ± 0.01
Lignoceric	C24:0		0.06 ± 0.05	0.13 ± 0.01
Nervonic	C24:1c15		0.01 ± 0.01	0.06 ± 0.02
Fatty acid ratios:				
Total saturated fatty acids (SFAs)			71.31 ± 2.58	80.24 ± 3.89
Total monounsaturated fatty acids (MUFAs)			24.62 ± 2.17	16.82 ± 3.37
Total polyunsaturated fatty acids (PUFAs)			4.08 ± 0.71	2.95 ± 0.64
Total omega-3 fatty acids			1.24 ± 0.20	2.21 ± 0.72
Total omega-6 fatty acids			2.83 ± 0.81	0.74 ± 0.18

n = number of milk samples; N = number of individual animals; ND = not detected.

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
