# Peer review of "Milk Composition of Free-Ranging Impala (Aepyceros melampus) and Tsessebe (Damaliscus lunatus lunatus), and Comparison with Other African Bovidae"

_animals, 2021, doi:10.3390/ani11020516_

Round 1

Reviewer 1 Report

Dear authors, 

After carefully reviewing the new version of Animals-1094708 R1, I must conclude that the manuscript has definitely improved. 

I still have some strong reservations about the reliability regarding the numbers used for the experimental design. However, you have appropriately explained in your letter the limitations of working with free range non-domesticated mammals. I certainly reckon that data regarding cattle and small ruminants is well established in literature, which may not be possible to this extent in the case of Impala and Tsessebe. 

However, I still believe there is scope for improvement and these limitations need to be properly clarified at some point in the text. Please consider including a paragraph addressing these limitations. 

See also the following minor details:

  1. As I commented before, please consider including in the title the latin names of the species after their common name: (Damaliscus lunatus lunatus and Aepyceros melampus).
  2. Throughout the manuscript, the authors refer to “lambs” and “calves” as the progeny of both Impala and Tsessebe. I am not sure if these terms are correct. Please check and use the same term in all the text. 

Thank you for considering my suggestions.

Kind regards.

Author Response

General response

We wish to thank the reviewers for their strict and constructive comments. We appreciate the input which improved our manuscript.

Reviewer 1.

*Thank you for the suggestion of adding a short description of limitations of studies with wild animals. We have not included this in previous work, and not seen such statements in other publications. The reason might be to save on writing space, or, more importantly, fear of jeopardizing the research data. It was difficult to find the right spot, but was included in the introduction at the end of the first paragraph.

*The latin names of the species were included in the title.

*According to reference 34, the family of the impala consists of ram, ewe and lamb, while that of the Alcelaphinae members (with the exception of blesbok and bontebok) of bull, cow and calf.

Reviewer 2 Report

The authors significantly improved the paper.

Author Response

General response

We wish to thank the reviewers for their strict and constructive comments. We appreciate the input which improved our manuscript.

Reviewer 1.

*Thank you for the suggestion of adding a short description of limitations of studies with wild animals. We have not included this in previous work, and not seen such statements in other publications. The reason might be to save on writing space, or, more importantly, fear of jeopardizing the research data. It was difficult to find the right spot, but was included in the introduction at the end of the first paragraph.

*The latin names of the species were included in the title.

*According to reference 34, the family of the impala consists of ram, ewe and lamb, while that of the Alcelaphinae members (with the exception of blesbok and bontebok) of bull, cow and calf.

Reviewer 2.

Thank you for reviewing.

Reviewer 3

Thank you for reviewing.

Reviewer 3 Report

The authors made the required changes and explained all the difficulties encountered in investigating wild animals, I think the manuscript is now ready for publication.

Author Response

General response

We wish to thank the reviewers for their strict and constructive comments. We appreciate the input which improved our manuscript.

Reviewer 1.

*Thank you for the suggestion of adding a short description of limitations of studies with wild animals. We have not included this in previous work, and not seen such statements in other publications. The reason might be to save on writing space, or, more importantly, fear of jeopardizing the research data. It was difficult to find the right spot, but was included in the introduction at the end of the first paragraph.

*The latin names of the species were included in the title.

*According to reference 34, the family of the impala consists of ram, ewe and lamb, while that of the Alcelaphinae members (with the exception of blesbok and bontebok) of bull, cow and calf.

Reviewer 2.

Thank you for reviewing.

Reviewer 3

Thank you for reviewing.

This manuscript is a resubmission of an earlier submission. The following is a list of the peer review reports and author responses from that submission.

Round 1

Reviewer 1 Report

Dear Authors, 

I have carefully reviewed the manuscript "Milk composition of free-ranging Impala and Tsessebe, and comparison with other African Bovidae”. Although the subject might be of interest for the field, I have found many issues in the experimental design and a considerable lack of information that should be provided in order to understand the results and establish comparisons with other species. 

I also strongly recommend the authors to seek a professional translator or a native English speaker to check the text prior to further submissions. 

Please find below detailed comments to the manuscript.

TITLE

Please include in the title the latin names of the species after their common name: (Damaliscus lunatus lunatus and Aepyceros melampus).

INTRODUCTION

There is no information whatsoever throughout the manuscript about the two species included in the study. I am convinced in order to understand what is being done it is essential to provide a thorough description of Tsessebe and Impala. Authors should include details about population, use of these species in farming, approximate census, etc., and also answer some of the following questions: Are these animals extensively farmed? How long does their lactation last? What kind of curves of lactation do they follow? For us readers it is absolutely imposible to understand the results without knowing what kind of animals we are dealing with.

MATERIAL AND METHODS

The experimental design has considerable flaws. I have deep concerns about the number of animals included in the study (as low as only 3 in the case of Tsessebe!). Not only the number of animals sampled is very limited, but also the source of these animals can considerably bias the results, as there could be a strong herd/flock effect that the authors are ignoring. In my opinion, this experimental design does not allow to establish any valid and reliable conclusions.

I am also concerned about the facts surrounding stage of lactation. Tsessebes and Impalas included in the study are in different stages of lactation, which can entail considerable variations in milk solids concentration. Therefore, results are not standardised enough to be compared not only with other species but also within the two studied subject to study.

Have the authors performed replicas of the analyses? I think it is unlikely due to the total volume of milk sampled from each animal. This should be at least clarified in the methodology.

L118-119? Why have the authors only included cow and sheep as a reference when identifying protein bands? I would strongly suggest including also goat and at least any other specie phylogenetically related to Tsessebe and Impala in order to run Urea-PAGE electrophoresis. 

RESULTS AND DISCUSSION 

As mentioned before, the experimental design does not allow to establish proper comparisons between species. 

For instance it is difficult to compare protein content with other ruminants as there might be fluctuations throughout the lactation period (L195-196). Impalas are described to be either around 10 days post partum and 3-7 weeks. Some of these animals could have been sampled before and after the peak of lactation, delivering different results (again, this is unknown to the reader, as no information about the species has been included in the introduction or other section). In the case of Tsessebe, the 3 animals have been sampled at approximately 4-6 months of lactation: would this be considered mid or late lactation? This information is unknown to the reader, who only knows that literature has proved variation in milk solids as a concentration effect due to lower yields as lactation progresses. In a similar way, the described lactose content in L220-221, could be higher because of this concentration effect due low yields in late lactation.

TABLE 1

The number n of impalas is stated as 20, although in the materials and methods section it is described as 10 (4+6). Please amend or clarify this. 

FIGURE 1 

Why did the authors include in the gel only 2 Impalas and all the 3 Tsessebes? Were the two lanes tagged as Impala individual samples or composite samples from each of the 2 herds? This needs to be clarified.

Reviewer 2 Report

I find the article Milk composition of free-ranging impala and tsessebe, and comparison with other African Bovidae very original and provides information about the composition of milk from wild African ruminants. I think it is necessary to incorporate information regarding the duration of lactation of both species. There is no analysis regarding the small number of samples that are analyzed and how the milk of these species can be compared in different stages of their lactation.

Reviewer 3 Report

Revision of manuscript entitled “Milk composition of free-ranging impala and tsessebe, and comparison with other African Bovidae”.
The manuscript is very interesting, anyway some issues need to be addressed.
In the introduction section, the authors describe well the state of the art, but it can be improved, ie the authors should state clearly why it is important to study the chemical composition of milk of these species, and what the purpose of their work is.
The statistical analysis is confused, as the author refer to previous publications (of the same authors) as part of the present manuscript. Again, the authors should make the manuscript itself readable, possibly by adding images or tables as supplementary files. In addition, in my opinion, statistical analysis should respect the manuscript sections (mat and met, result, discussion), ie I would have preferred to find the differences of fatty acid composition described in the results section and discussed in the discussion section, perhaps with a table explaining the significant and non-significant differences, even in a supplementary file. The way statistical analysis is organized is very confusing.

Line 22: please remove the hyphen in "nutrient-", and throughout the text (except the title).
Line 73: lactose is also the main osmole of milk and its production is associated with the movement of water into the secretory vescicle (Bleck and Bremel, 1993). Bleck GT & Bremel RD 1993 Correlation of the α-lactalbumin (+15) polymorphism to milk production and milk composition of Holsteins. Journal of Dairy Science 76 2292–2298
Line 80-84, please rephrase or explain, you are referring to your own research path, or your laboratory's own research path, which the reader is not required to reconstruct. You should specify the publications you are referring to, whether they are yours or of other authors, it makes no difference.
Line 85-88: the purpose of the work is not entirely clear, it could be better formulated.
line 96: how many tsetssebi did you milked?
Lines 172-173: “The milk fat and -oligosaccharide content of the species under study are in the same order of magnitude as in cow’s milk, it was assumed that the analytical techniques were not affected to the same extent.” Please rephrase, the sentence is not clear, furthermore, it better fits to the discussion section.
Line 178: you must also indicate the meaning of acronyms NPN and FFDM in the caption of the Table.
Line 183: electropherograms
Line 251-252: it has been shown that the diet affects milk fatty acid composition, perhaps it does not affect the differences observed between animals grazing in the same territory.
Line 259: “GE” please specify each acronim the first time it appears in the text.
Line 279-280: “No refinement of the 2017 report could be demonstrated” please cite the literature properly, the reader must be able to understand the present manuscript without having all the previous publications.